# Saliency Diversified Deep Ensemble for Robustness to Adversaries

## Alex Bogun, Dimche Kostadinov, Damian Borth

University of St. Gallen
alex.bogun@unisg.ch, dimche.kostadinov@unisg.ch, damian.borth@unisg.ch

## Abstract

Deep learning models have shown incredible performance on numerous image recognition, classification, and reconstruction tasks. Although very appealing and valuable due to their predictive capabilities, one common threat remains challenging to resolve. A specifically trained attacker can introduce malicious input perturbations to fool the network, thus causing potentially harmful mispredictions. Moreover, these attacks can succeed when the adversary has full access to the target model (white-box) and even when such access is limited (black-box setting). The ensemble of models can protect against such attacks but might be brittle under shared vulnerabilities in its members (attack transferability). To that end, this work proposes a novel diversity-promoting learning approach for the deep ensembles. The idea is to promote saliency map diversity (SMD) on ensemble members to prevent the attacker from targeting all ensemble members at once by introducing an additional term in our learning objective. During training, this helps us minimize the alignment between model saliencies to reduce shared member vulnerabilities and, thus, increase ensemble robustness to adversaries. We empirically show a reduced transferability between ensemble members and improved performance compared to the state-of-the-art ensemble defense against medium and high-strength white-box attacks. In addition, we demonstrate that our approach combined with existing methods outperforms state-of-the-art ensemble algorithms for defense under white-box and black-box attacks.

## 1  Introduction

Nowadays, deep learning models have shown incredible performance on numerous image recognition, classification, and reconstruction tasks (Krizhevsky, Sutskever, and Hinton 2012; Lee et al. 2015; LeCun, Bengio, and Hinton 2015; Chen et al. 2020). Due to their great predictive capabilities, they have found widespread use across many domains (Szegedy et al. 2016; Devlin et al. 2019; Deng, Hinton, and Kingsbury 2013). Although deep learning models are very appealing for many interesting tasks, their robustness to adversarial attacks remains a challenging problem to solve. A specifically trained attacker can introduce malicious input perturbations to fool the network, thus causing potentially harmful (Goodfellow, Shlens, and Szegedy 2015; Madry

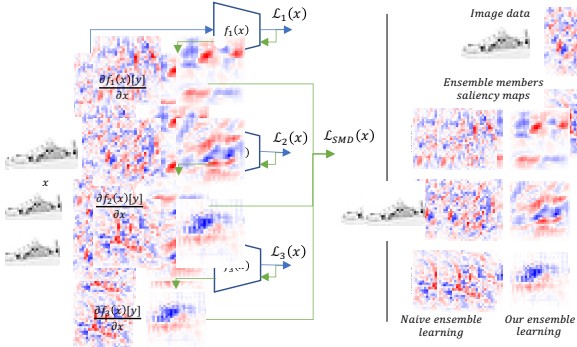

Figure 1: **Left.** An illustration of the proposed learning scheme for saliency-based diversification of deep ensemble consisting of 3 members. We use the cross-entropy losses $\mathcal{L}_m(x), m \in \{1, 2, 3\}$ and regularization $\mathcal{L}_{SMD}(x)$ for saliency-based diversification. **Right.** An example of saliency maps for members of naively learned ensemble and learned ensemble with our approach. Red and blue pixels represent positive and negative saliency values respectively.

et al. 2018) mispredictions. Moreover, these attacks can succeed when the adversary has full access to the target model (white-box) (Athalye and Carlini 2018) and even when such access is limited (black-box) (Papernot et al. 2017), posing a hurdle in security- and trust-sensitive application domains.

The ensemble of deep models can offer protection against such attacks (Strauss et al. 2018). Commonly, an ensemble of models has proven to improve the robustness, reduce variance, increase prediction accuracy and enhance generalization compared to the individual models (LeCun, Bengio, and Hinton 2015). As such, ensembles were offered as a solution in many areas, including weather prediction (Palmer 2019), computer vision (Krizhevsky, Sutskever, and Hinton 2012), robotics and autonomous driving (Kober, Bagnell, and Peters 2013) as well as others, such as (Ganaie et al. 2021). However, 'naive' ensemble models are brittle due to shared vulnerabilities in their members (Szegedy et al. 2016). Thus an adversary can exploit attack *transferability* (Madry et al. 2018) to affect all members and the ensemble as a whole.

In recent years, researchers tried to improve the adversarial robustness of the ensemble by maximizing different no-

tions for diversity between individual networks (Pang et al. 2019; Kariyappa and Qureshi 2019; Yang et al. 2020). In this way, adversarial attacks that fool one network are much less likely to fool the ensemble as a whole (Chen et al. 2019b; Sen, Ravindran, and Raghunathan 2019; Tramèr et al. 2018; Zhang, Liu, and Yan 2020). The research focusing on ensemble diversity aims to diversely train the neural networks inside the ensemble model to withstand the deterioration caused by adversarial attacks. The works (Pang et al. 2019; Zhang, Liu, and Yan 2020; Kariyappa and Qureshi 2019) proposed improving the diversity of the ensemble constituents by training the model with diversity regularization in addition to the main learning objective. (Kariyappa and Qureshi 2019) showed that an ensemble of models with misaligned loss gradients can be used as a defense against black-box attacks and proposed uncorrelated loss functions for ensemble learning. (Pang et al. 2019) proposed an adaptive diversity promoting (ADP) regularizer to encourage diversity between non-maximal predictions. (Yang et al. 2020) minimize vulnerability diversification objective in order to suppress shared 'week' features across the ensemble members. However, some of these approaches only focused on white-box attacks (Pang et al. 2019), black-box attacks (Kariyappa and Qureshi 2019) or were evaluated on a single dataset (Yang et al. 2020).

In this paper, we propose a novel diversity-promoting learning approach for deep ensembles. The idea is to promote Saliency Map Diversity (SMD) to prevent the attacker from targeting all ensemble members at once.

Saliency maps (SM) (Gu and Tresp 2019) represent the derivative of the network prediction for the actual true label with respect to the input image. They indicate the most 'sensitive' content of the image for prediction. Intuitively, we would like to learn an ensemble whose members have different sensitivity across the image content while not sacrificing the ensemble predictive power. Therefore, we introduce a *saliency map diversity (SMD)* regularization term in our learning objective. Given image data and an ensemble of models, we define the SMD using the inner products between all pairs of saliency maps (for one image data, one ensemble member has one saliency map). Different from our approach with SMD regularization, (Pang et al. 2019) defined the diversity measure using the non-maximal predictions of individual members, and as such might not be able to capture the possible shared sensitivity with respect to the image content related to the correct predictions.

We jointly learn our ensemble members using cross-entropy losses (LeCun, Bengio, and Hinton 2015) for each member and our shared *SMD* term. This helps us minimize the alignment between model SMDs and enforces the ensemble members to have misaligned and non-overlapping sensitivity for the different image content. Thus with our approach, we try to minimize possible shared sensitivity across the ensemble members that might be exploited as vulnerability, which is in contrast to (Yang et al. 2020) who try to minimize shared 'week' features across the ensemble members. It is also important to note that our regularization differs from (Kariyappa and Qureshi 2019), since it focuses on gradients coming from the correct class predictions (saliencies),

which could also be seen as a loss agnostic approach. We illustrate our learning scheme in Fig. 1, left. Whereas in Fig. 1 on the right, we visualize the saliency maps with respect to one image sample for the members in naively trained ensemble and an ensemble trained with our approach.

We perform an extensive numerical evaluation using the MNIST (Lecun et al. 1998), Fashion-MNIST (F-MNIST) (Xiao, Rasul, and Vollgraf 2017), and CIFAR-10 (Krizhevsky 2009) datasets to validate our approach. We use two neural networks architectures and conduct experiments for different known attacks and at different attack strengths. Our results show a reduced transferability between ensemble members and improved performance compared to the state-of-the-art ensemble defense against medium and high-strength white-box attacks. Since we minimize the shared sensitivity which could also be seen as the attention of a prediction important image content, we also suspected that our approach could go well with other existing methods. To that end, we show that our approach combined with the (Yang et al. 2020) method outperforms state-of-the-art ensemble algorithms for defense under adversarial attacks in both white-box and black-box settings. We summarize our main contributions in the following:

- We propose a diversity-promoting learning approach for deep ensemble, where we introduce a saliency-based regularization that diversifies the sensitivity of ensemble members with respect to the image content.

- We show improved performance compared to the state-of-the-art ensemble defense against medium and high strength white-box attacks as well as show on-pair performance for the black-box attacks.

- We demonstrate that our approach combined with the (Yang et al. 2020) method outperforms state-of-the-art ensemble defense algorithms in white-box and black-box attacks.

## 2 Related Work

In this section, we overview the recent related work.

### 2.1 Common Defense Strategies

In the following, we describe the common defense strategies against adversarial attacks groping them into four categories.

**Adversarial Detection.** These methods aim to detect the adversarial examples or to restore the adversarial input to be closer to the original image space. Adversarial Detection methods (Bhambri et al. 2020) include *MagNet*, *Feature Squeezing*, and *Convex Adversarial Polytope*. The *MagNet* (Meng and Chen 2017) method consists of two parts: detector and reformer. Detector aims to recognize and reject adversarial images. Reformer aims to reconstruct the image as closely as possible to the original image using an auto-encoder. The *Feature Squeezing* (Xu, Evans, and Qi 2018) utilizes feature transformation techniques such as squeezing color bits and spatial smoothing. These methods might be prone to reject clean examples and might have to severely modify the input to the model. This could reduce the performance on the clean data.

**Gradient Masking and Randomization Defenses.** Gradient masking represents manipulation techniques that try to hide the gradient of the network model to robustify against attacks made with gradient direction techniques and includes distillation, obfuscation, shattering, use of stochastic and vanishing or exploding gradients (Papernot et al. 2017; Athalye, Carlini, and Wagner 2018; Carlini and Wagner 2017). The authors in (Papernot et al. 2016b) introduced a method based on *distillation*. It uses an additional neural network to 'distill' labels for the original neural network in order to reduce the perturbations due to adversarial samples. (Xie et al. 2018) used a *randomization* method during training that consists of random resizing and random padding for the training image data. Another example of such randomization can be noise addition at different levels of the system (You et al. 2019), injection of different types of randomization like, for example, random image resizing or padding (Xie et al. 2018) or randomized lossy compression (Das et al. 2018), etc. As a disadvantage, these approaches can reduce the accuracy since they may reduce useful information, which might also introduce instabilities during learning. As such, it was shown that often they can be easily bypassed by the adversary via expectation over transformation techniques (Athalye and Carlini 2018).

**Secrecy-based Defenses.** The third group generalizes the defense mechanisms, which include randomization explicitly based on a secret key that is shared between training and testing stages. Notable examples are random projections (Vinh et al. 2016), random feature sampling (Chen et al. 2019a) and the key-based transformation (Taran, Rezaeifar, and Voloshynovskiy 2018), etc. As an example in (Taran et al. 2019) introduces randomized diversification in a special transform domain based on a secret key, which creates an information advantage to the defender. Nevertheless, the main disadvantage of the known methods in this group consists of the loss of performance due to the reduction of useful data that should be compensated by a proper diversification and corresponding aggregation with the required secret key.

**Adversarial Training (AT).** (Goodfellow, Shlens, and Szegedy 2015; Madry et al. 2018) proposed one of the most common approaches to improve adversarial robustness. The main idea is to train neural networks on both clean and adversarial samples and force them to correctly classify such examples. The disadvantage of this approach is that it can significantly increase the training time and can reduce the model accuracy on the unaltered data (Tsipras et al. 2018).

## 2.2 Diversifying Ensemble Training Strategies

Even naively learned ensemble could add improvement towards adversarial robustness. Unfortunately, ensemble members may share a large portion of vulnerabilities (Dauphin et al. 2014) and do not provide any guarantees to adversarial robustness (Tramèr et al. 2018).

(Tramèr et al. 2018) proposed Ensemble Adversarial Training (*EAT*) procedure. The main idea of EAT is to minimize the classification error against an adversary that maximizes the error (which also represents a min-max optimization problem (Madry et al. 2018)). However, this approach is very computationally expensive and according to the original author may be vulnerable to white-box attacks.

Recently, diversifying the models inside an ensemble gained attention. Such approaches include a mechanism in the learning procedure that tries to minimize the adversarial subspace by making the ensemble members diverse and making the members less prone to shared weakness.

(Pang et al. 2019) introduced **ADP** regularizer to diversify training of the ensemble model to increase adversarial robustness. To do so, they defined first an Ensemble Diversity $ED = \text{Vol}^2(||f_m^{\backslash y}(x)||_2)$, where $f_m^{\backslash y}(x)$ is the order preserving prediction of $m$-th ensemble member on $x$ without $y$-th (maximal) element and $\text{Vol}(\cdot)$ is a total volume of vectors span. The ADP regularizer is calculated as $\text{ADP}_{\alpha,\beta}(x,y) = \alpha \cdot \mathcal{H}(\mathcal{F}) + \beta \cdot \log(ED)$, where $\mathcal{H}(\mathcal{F}) = -\sum_i f_i(x)\log(f_i(x))$ is a Shannon entropy and $\alpha, \beta > 0$. The ADP regularizer is then subtracted from the original loss during training.

The **GAL** regularizer (Kariyappa and Qureshi 2019) was intended to diversify the adversarial subspaces and reduce the overlap between the networks inside ensemble model. GAL is calculated using the cosine similarity (CS) between the gradients of two different models as $CS(\nabla_x \mathcal{J}_a, \nabla_x \mathcal{J}_b)_{a \neq b} = \frac{\langle \nabla_x \mathcal{J}_a, \nabla_x \mathcal{J}_b \rangle}{|\nabla_x \mathcal{J}_a| \cdot |\nabla_x \mathcal{J}_b|}$, where $\nabla_x \mathcal{J}_m$ is the gradient of the loss of $m$-th member with respect to x. During training, the authors added the term $GAL = \log\left(\sum_{1 \leq a < b \leq N} \exp(CS(\nabla_x \mathcal{J}_a, \nabla_x \mathcal{J}_b))\right)$ to the learning objective.

With **DVERGE** (Yang et al. 2020), the authors aimed to maximize the vulnerability diversity together with the original loss. They defined a *vulnerability diversity* between pairs of ensemble members $f_a(x)$ and $f_b(x)$ using data consisting of the original data sample and its *feature distilled* version. In other words, they deploy an ensemble learning procedure where each ensemble member $f_a(x)$ is trained using adversarial samples generated by other members $f_b(x)$, $a \neq b$.

## 2.3 Adversarial Attacks

The goal of the adversary is to craft an image $x'$ that is very close to the original $x$ and would be correctly classified by humans but would fool the target model. Commonly, attackers can act as adversaries in white-box and black-box modes, depending on the gained access level over the target model.

**White-box and Black-box Attacks.** In the white-box scenario, the attacker is fully aware of the target model's architecture and parameters and has access to the model's gradients. White-box attacks are very effective against the target model but they are bound to the extent of knowing the model. In the Black-box scenario, the adversary does not have access to the model parameters and may only know the training dataset and the architecture of the model (in grey-box setting). The attacks are crafted on a surrogate model but still work to some extent on the target due to transferability (Papernot et al. 2016a).

An adversary can build a white-box or black-box attack using different approaches. In the following text, we briefly

describe the methods commonly used for adversarial attacks.

**Fast Gradient Sign Method (FGSM).** (Goodfellow, Shlens, and Szegedy 2015) generated adversarial attack $x'$ by adding the sign of the gradient $\text{sign}(\nabla_x \mathcal{J}(x, y))$ as perturbation with $\epsilon$ strength, *i.e.*, $x' = x + \epsilon \cdot \text{sign}(\nabla_x \mathcal{J}(x, y))$.

**Random Step-FGSM (R-FGSM).** The method proposed in (Tramèr et al. 2018) is an extension of FGSM where a single random step is taken before FGSM due to the assumed non-smooth loss function in the neighborhood of data points.

**Projected Gradient Descent (PGD).** (Madry et al. 2018) presented a similar attack to BIM, with the difference that they randomly selected the initialization of $x'_0$ in a neighborhood $\dot{U}(x, \epsilon)$.

**Basic Iterative Method (BIM).** (Kurakin, Goodfellow, and Bengio 2017) proposed iterative computations of attack gradient for each smaller step. Thus, generating an attacks as $x'_i = \text{clip}_{x, \epsilon}(x'_{i-1} + \frac{\epsilon}{r} \cdot \text{sign}(g_{i-1}))$, where $g_i = \nabla_x \mathcal{J}(x'_i, y)$, $x'_0 = x$ and $r$ is the number of iterations.

**Momentum Iterative Method (MIM).** (Dong et al. 2018) proposed extenuation of BIM. It proposes to update gradient with the momentum $\mu$ to ensure best local minima. Holding the momentum helps to avoid small holes and poor local minimum solution, $g_i = \mu g_{i-1} + \frac{\nabla_x \mathcal{J}(x'_{i-1}, y)}{||\nabla_x \mathcal{J}(x'_{i-1}, y)||_1}$.

## 3 Saliency Diversified Ensemble Learning

In this section, we present our diversity-promoting learning approach for deep ensembles. In the first subsection, we introduce the saliency-based regularizer, while in the second subsection we describe our learning objective.

### 3.1 Saliency Diversification Measure

**Saliency Map.** In (Etmann et al. 2019), the authors investigated the connection between a neural network's robustness to adversarial attacks and the interpretability of the resulting saliency maps. They hypothesized that the increase in interpretability could be due to a higher alignment between the image and its saliency map. Moreover, they arrived at the conclusion that the strength of this connection is strongly linked to how locally similar the network is to a linear model. In (Mangla, Singh, and Balasubramanian 2020) authors showed that using weak saliency maps suffices to improve adversarial robustness with no additional effort to generate the perturbations themselves.

We build our approach on prior work about saliency maps and adversarial robustness but in the context of deep ensemble models. In (Mangla, Singh, and Balasubramanian 2020) the authors try to decrease the sensitivity of the prediction with respect to the saliency map by using special augmentation during training. We also try to decrease the sensitivity of the prediction with respect to the saliency maps but for the ensemble. We do so by enforcing misalignment between the saliency maps for the ensemble members.

We consider a saliency map for model $f_m$ with respect to data $x$ conditioned on the true class label $y$. We calculate it

as the first order derivative of the model output for the true class label with respect to the input, *i.e.*,

$$s_m = \frac{\partial f_m(x)[y]}{\partial x}, \qquad (1)$$

where $f_m(x)[y]$ is the $y$ element from the predictions $f_m(x)$.

**Shared Sensitivity Across Ensemble Members.** Given image data $x$ and an ensemble of $M$ models $f_m$, we define our SMD measure as:

$$\mathcal{L}_{SMD}(x) = \log\left[\sum_m \sum_{l>m} \exp\left(\frac{s_m^T s_l}{\|s_m\|_2 \|s_l\|_2}\right)\right], \quad (2)$$

where $s_m = \frac{\partial f_m(x)[y]}{\partial x}$ is the saliency map for ensemble model $f_m$ with respect to the image data $x$. A high value of $\mathcal{L}_{SMD}(x)$ means alignment and similarity between the saliency maps $s_m$ of the models $f_m(x)$ with respect to the image data $x$. Thus SMD (2) indicates a possible shared sensitivity area in the particular image content common for all the ensemble members. A pronounced sensitivity across the ensemble members points to a vulnerability that might be targeted and exploited by an adversarial attack. To prevent this, we would like $\mathcal{L}_{SMD}(x)$ to be as small as possible, which means different image content is of different importance to the ensemble members.

### 3.2 Saliency Diversification Objective

We jointly learn our ensemble members using a common cross-entropy loss per member and our saliency based sensitivity measure described in the subsection above. We define our learning objective in the following:

$$\mathcal{L} = \sum_x \sum_m \mathcal{L}_m(x) + \lambda \sum_x \mathcal{L}_{SMD}(x), \qquad (3)$$

where $\mathcal{L}_m(x)$ is the cross-entropy loss for ensemble member $m$, $\mathcal{L}_{SMD}(x)$ is our SMD measure for an image data $x$ and an ensemble of $M$ models $f_m$, and $\lambda > 0$ is a Lagrangian parameter. By minimizing our learning objective that includes a saliency-based sensitivity measure, we enforce the ensemble members to have misaligned and non-overlapping sensitivity for the different image content. Our regularization enables us to strongly penalize small misalignments $s_m^T s_l$ between the saliency maps $s_m$ and $s_l$. While at the same time it ensures that a large misalignment is not discarded. Additionally, since $\mathcal{L}_{SMD}(x)$ is a $logSumExp$ function it has good numerical properties (Kariyappa and Qureshi 2019). Thus, our approach offers to effectively minimize possible shared sensitivity across the ensemble members that might be exploited as vulnerability. In contrast to GAL regularizer (Kariyappa and Qureshi 2019) SMD is loss agnostic (can be used with loss functions other than cross-entropy) and does not focus on incorrect-class prediction (which are irrelevant for accuracy). Additionally it has a clear link to work in interpretability (Etmann et al. 2019) and produces diverse but meaningful saliency maps (see Fig. 1).

Assuming unit one norm saliencies, the gradient based update for one data sample $x$ with respect to the parameters

$\theta_{f_m}$ of a particular ensemble member can be written as:

$$\theta_{f_m} = \theta_{f_m} - \alpha\left(\frac{\partial \mathcal{L}_m(x)}{\partial \theta_{f_m}} + \lambda \frac{\partial \mathcal{L}_{SMD}(x)}{\partial \theta_{f_m}}\right) =$$
$$= \theta_{f_m} - \alpha \frac{\partial \mathcal{L}_m(x)}{\partial \theta_{f_m}} - \alpha\lambda \frac{\partial f_m(x)[y]}{\partial x \partial \theta_{f_m}} \sum_{j \neq m} \beta_j \frac{\partial f_j(x)[y]}{\partial x}, \quad (4)$$

where $\alpha$ is the learning rate and $\beta_j = \frac{\exp(s_m^T s_j)}{\sum_m \sum_{k>m} \exp(s_m^T s_k)}$. The third term enforces the learning of the ensemble members to be on optimization paths where the gradient of their saliency maps $\frac{\partial f_m(x)[y]}{\partial x \partial \theta_{f_m}}$ with respect to $\theta_{f_m}$ is misaligned with the weighted average of the remaining saliency maps $\sum_{j \neq m} \beta_j \frac{\partial f_j(x)[y]}{\partial x}$. Also, (4) reveals that by our approach the ensemble members can be learned in parallel provided that the saliency maps are shared between the models (we leave this direction for future work).

## 4    Empirical Evaluation

This section is devoted to empirical evaluation and performance comparison with state-of-the-art ensemble methods.

### 4.1    Data Sets and Baselines

We performed the evaluation using 3 classical computer vision data sets (MNIST (Lecun et al. 1998), FASHION-MNIST (Xiao, Rasul, and Vollgraf 2017) and CIFAR-10 (Krizhevsky 2009)) and include 4 baselines (naive ensemble, (Pang et al. 2019), (Kariyappa and Qureshi 2019), (Yang et al. 2020)) in our comparison.

**Datasets.**    The MNIST dataset (Lecun et al. 1998) consists of 70000 gray-scale images of handwritten digits with dimensions of 28x28 pixels. F-MNIST dataset (Xiao, Rasul, and Vollgraf 2017) is similar to MNIST dataset, has the same number of images and classes. Each image is in grayscale and has a size of 28x28. It is widely used as an alternative to MNIST in evaluating machine learning models. CIFAR10 dataset (Krizhevsky 2009) contains 60000 color images with 3 channels. It includes 10 real-life classes. Each of the 3 color channels has a dimension of 32x32.

**Baselines.**    As the simplest baseline we compare against the performance of a naive ensemble, *i.e.*, one trained without any defense mechanism against adversarial attacks. Additionally, we also consider state-of-the-art methods as baselines. We compare the performance of our approach with the following ones: Adaptive Diversity Promoting (ADP) method (Pang et al. 2019), Gradient Alignment Loss (GAL) method (Kariyappa and Qureshi 2019), and a Diversifying Vulnerabilities for Enhanced Robust Generation of Ensembles (DVERGE) or (DV.) method (Yang et al. 2020).

### 4.2    Training and Testing Setup

**Used Neural Networks.**    To evaluate our approach, we use two neural networks LeNet-5 (Lecun et al. 1998) and ResNet-20 (He et al. 2016). LeNet-5 is a classical small neural network for vision tasks, while ResNet-20 is another widely used architecture in this domain.

**Training Setup.**    We run our training algorithm for 50 epochs on MNIST and F-MNIST and 200 epochs on CIFAR-10, using the Adam optimizer (Kingma and Ba 2015), a learning rate of 0.001, weight decay of 0.0001, and batch-sizes of 128. We use no data augmentation on MNIST and F-MNIST and use normalization, random cropping, and flipping on CIFAR-10. In all of our experiments, we use 86% of the data for training and 14% for testing. In the implemented regularizers from prior work, we used the $\lambda$ that was suggested by the respective authors. While we found out that the strength of the SMD regularizer (also $\lambda$) in the range $[0.5, 2]$ gives good results. Thus in all of our experiments, we take $\lambda = 1$. We report all the results as an average over 5 independent trials (we include the standard deviations in the Appendix A). We report results for the ensembles of 3 members in the main paper, and for 5 and 8 in the Appendix C.

We used the LeNet-5 neural network for MNIST and F-MNIST datasets and ResNet-20 for CIFAR-10. To have a fair comparison, we also train ADP (Pang et al. 2019), GAL (Kariyappa and Qureshi 2019) and DVERGE (Yang et al. 2020), under a similar training setup as described above. We made sure that the setup is consistent with the one given by the original authors with exception of using Adam optimizer for training DVERGE. We also used our approach and added it as a regularizer to the DVERGE algorithm. We named this combination SMD+ and ran it under the setup as described above. All models are implemented in PyTorch (Paszke et al. 2017). We use AdverTorch (Ding, Wang, and Jin 2019) library for adversarial attacks.

In the setting of adversarial training, we follow the EAT approach (Tramèr et al. 2018) by creating adversarial examples on 3 holdout pre-trained ensembles with the same size and architecture as the baseline ensemble. The examples are created via PGD-$L_\infty$ attack with 10 steps and $\epsilon = 0.1$.

**Adversarial Attacks.**    To evaluate our proposed approach and compare its performance to baselines, we use a set of adversarial attacks described in Section 2.3 in both black-box and white-box settings. We construct adversarial examples from the images in the test dataset by modifying them using the respective attack method. We probe with white-box attacks on the ensemble as a whole (not on the individual models). We generate black-box attacks targeting our ensemble model by creating white-box adversarial attacks on a surrogate ensemble model (with the same architecture), trained on the same dataset with the same training routine. We use the following parameters for the attacks: for ($F_{GSM}$, PGD, R-F., BIM, MIM) we use $\epsilon$ in range $[0; 0.3]$ in 0.05 steps, which covers the range used in our baselines; we use 10 iterations with a step size equal to $\epsilon/10$ for PGD, BIM and MIM; we use $L_\infty$ variant of PGD attack; for R-F. we use random-step $\alpha = \epsilon/2$.

**Computing Infrastructure and Run Time.**    As computing hardware, we use half of the available resources from NVIDIA DGX2 station with 3.3GHz CPU and 1.5TB RAM memory, which has a total of 16 1.75GHz GPUs, each with 32GB memory. One experiment takes around 4 minutes to train the baseline ensemble of 3 LeNet-5 members on

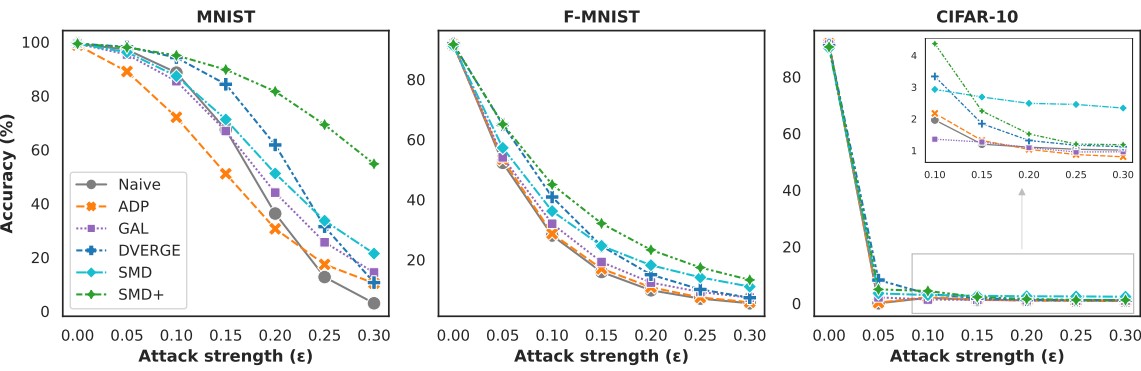

Figure 2: Accuracy vs. attacks strength for white-box PGD attacks on an ensemble of 3 LeNet-5 models for MNIST and F-MNIST and on an ensemble of 3 ReNets-20 for CIFAR-10 dataset.

| | MNIST | | | | | | F-MNIST | | | | | | CIFAR-10 | | | | | |
| | Clean | $F_{gsm}$ | R-F. | PGD | BIM | MIM | Clean | $F_{gsm}$ | R-F. | PGD | BIM | MIM | Clean | $F_{gsm}$ | R-F. | PGD | BIM | MIM |
|---|---|---|---|---|---|---|---|---|---|---|---|---|---|---|---|---|---|---|
| Naive | 99.3 | 20.3 | 73.5 | 2.9 | 4.2 | 5.5 | **91.9** | 15.7 | 33.6 | 5.5 | 7.2 | 6.6 | 91.4 | 10.5 | 2.8 | 1.0 | 3.2 | 2.9 |
| ADP | 98.8 | 43.8 | 89.6 | 10.4 | 19.6 | 14.8 | 91.4 | 18.3 | 34.8 | 5.8 | 8.8 | 7.5 | **91.7** | 11.4 | 3.7 | 0.8 | 3.6 | 3.4 |
| GAL | 99.3 | 72.7 | 89.0 | 14.4 | 28.2 | 38.9 | 91.4 | 35.8 | 51.2 | 7.4 | 10.8 | 12.2 | 91.4 | 11.2 | 9.7 | 1.0 | 1.8 | 2.8 |
| DV. | **99.4** | 44.2 | 85.5 | 10.6 | 16.0 | 20.6 | 91.8 | 27.3 | 44.6 | 7.3 | 10.7 | 9.9 | 91.0 | 11.2 | 6.3 | 1.1 | 5.5 | 4.4 |
| SMD | 99.3 | 70.7 | 91.3 | 21.4 | 34.3 | 43.8 | 91.1 | 38.2 | **52.0** | 11.0 | 14.9 | 16.4 | 90.1 | 12.0 | **12.0** | **2.3** | 3.2 | 3.9 |
| SMD+ | **99.4** | **83.4** | **93.8** | **54.7** | **68.0** | **71.0** | 91.6 | **42.9** | 51.9 | **13.3** | **20.5** | **20.5** | 90.5 | **12.1** | 5.8 | 1.2 | **5.9** | **5.2** |

Table 1: White-box attacks of the magnitude $\epsilon = 0.3$ on an ensemble of 3 LeNet-5 models for MNIST and F-MNIST and on an ensemble of 3 ReNets-20 for CIFAR-10 dataset. Columns are attacks and rows are defenses employed.

MNIST without any regularizer. Whereas it takes around 18 minutes to train the same ensemble under the SMD regularizer, 37 minutes under DVERGE regularize, and 48 minutes under their combination. To evaluate the same ensemble under all of the adversarial attacks takes approximately 1 hour. It takes approximately 3 days when ResNet-20 members are used on CIFAR-10 for the same experiment.

### 4.3 Results

**Robustness to White-Box Adversarial Attacks.** In Table 1, we show the results for ensemble robustness under white-box adversarial attacks with $\epsilon = 0.3$. We highlight in bold, the methods with the highest accuracy. In Figure 2, we depict the results for PGD attack at different attack strengths ($\epsilon$). It can be observed that the accuracy on normal images (without adversarial attacks) slightly decreases for all regularizers, which is consistent with a robustness-accuracy trade-off (Tsipras et al. 2018; Zhang et al. 2019). The proposed SMD and SMD+ outperform the comparing baselines methods on all attack configurations and datasets. This result shows that the proposed saliency diversification approach helps to increase the adversarial robustness.

**Robustness to Black-Box Adversarial Attacks.** In Table 2, we see the results for ensemble robustness under black-box adversarial attacks with an attack strength $\epsilon = 0.3$. In Figure 3 we also depict the results for PGD attack at different strengths ($\epsilon$). We can see that SMD+ is on par with DVERGE (DV.) on MNIST and consistently outper-

forms other methods. On F-MNIST SMD+ has a significant gap in performance compared to the baselines, with this effect being even more pronounced on the CIFAR-10 dataset. Also, it is interesting to note that standalone SMD comes second in performance and it is very close to the highest accuracy on multiple attack configurations under $\epsilon = 0.3$.

**Transferability.** In this subsection, we investigate the transferability of the attacks between the ensemble members, which measures how likely the crafted white-box attack for one ensemble member succeeds on another. In Figure 5, we present results for F-MNIST and PGD attacks (results for different datasets and other attacks are in the Appendix B). The Y-axis represents the member from which the adversary crafts the attack (i.e. source), and the X-axis - the member on which the adversary transfers the attack (i.e. target). The on diagonal values depict the accuracy of a particular ensemble member under a white-box attack. The other (off-diagonal) values show the accuracy of the target members under transferred (black-box) attacks from the source member. In Figure 5, we see that SMD and SMD+ have high ensemble resilience. It seems that both SMD and SMD+ reduce the common attack vector between the members. Compared to the naive ensemble and the DV. method, we see improved performance, showing that our approach increases the robustness to transfer attacks.

**Robustness Under Adversarial Training.** We also present the performance of our method and the comparing methods under AT. We follow the approach of Tramèr et al.

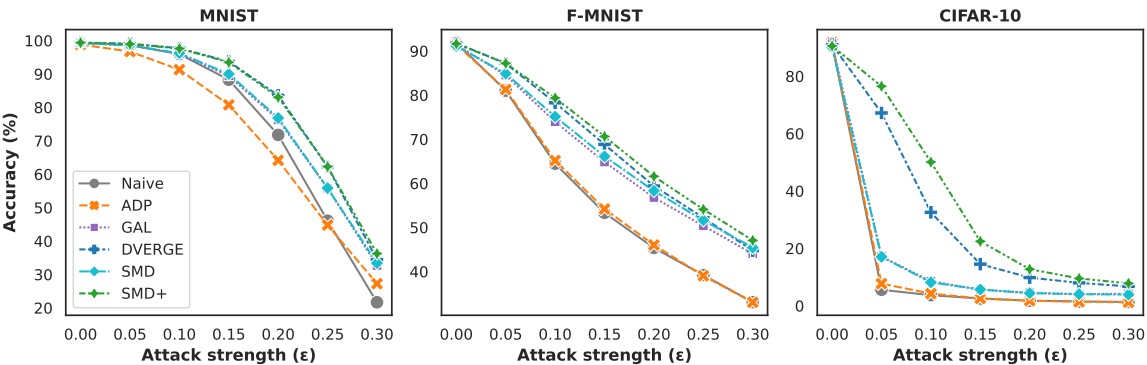

Figure 3: Accuracy vs. attacks strength for black-box PGD attacks on an ensemble of 3 LeNet-5 models for MNIST and F-MNIST and on an ensemble of 3 ReNets-20 for CIFAR-10 dataset.

|  | MNIST | | | | | | F-MNIST | | | | | | CIFAR-10 | | | | | |
| --- | --- | --- | --- | --- | --- | --- | --- | --- | --- | --- | --- | --- | --- | --- | --- | --- | --- | --- |
|  | Clean | $F_{gsm}$ | R-F. | PGD | BIM | MIM | Clean | $F_{gsm}$ | R-F. | PGD | BIM | MIM | Clean | $F_{gsm}$ | R-F. | PGD | BIM | MIM |
| Naive | 99.3 | 32.2 | 84.2 | 21.7 | 20.7 | 14.5 | 91.9 | 23.8 | 47.5 | 33.1 | 31.5 | 15.2 | 91.4 | 10.6 | 5.8 | 1.3 | 3.7 | 3.3 |
| ADP | 98.8 | 26.6 | 70.9 | 27.3 | 26.5 | 19.4 | 91.4 | 22.3 | 49.5 | 33.0 | 33.2 | 16.3 | **91.7** | **11.6** | 5.5 | 1.2 | 3.8 | 3.4 |
| GAL | 99.3 | 38.5 | 85.2 | 32.7 | 31.2 | 22.3 | 91.4 | 29.8 | 55.5 | 44.0 | 41.4 | 21.9 | 91.4 | 11.0 | 8.3 | 4.2 | 3.8 | **4.4** |
| DV. | **99.4** | **42.2** | **89.1** | 34.5 | 32.2 | 22.0 | 91.8 | 30.7 | 55.7 | 44.7 | 42.3 | 21.4 | 91.0 | 10.1 | 8.4 | 6.8 | 5.8 | 4.0 |
| SMD | 99.3 | 38.6 | 85.8 | 33.4 | 31.6 | 22.6 | 91.1 | 31.0 | 56.8 | 45.4 | 42.4 | 23.2 | 90.1 | 10.4 | 7.8 | 3.9 | 3.8 | 3.5 |
| SMD+ | **99.4** | 42.0 | **89.1** | **36.3** | **34.7** | **24.3** | 91.6 | **31.9** | **57.7** | **47.1** | **44.4** | **23.3** | 90.5 | 9.9 | **8.7** | **7.8** | **8.6** | 4.1 |

Table 2: Black-box attacks of the magnitude $\epsilon = 0.3$ on an ensemble of 3 LeNet-5 models for MNIST and F-MNIST and on an ensemble of 3 ReNets-20 for CIFAR-10 dataset. Columns are attacks and rows are defenses employed.

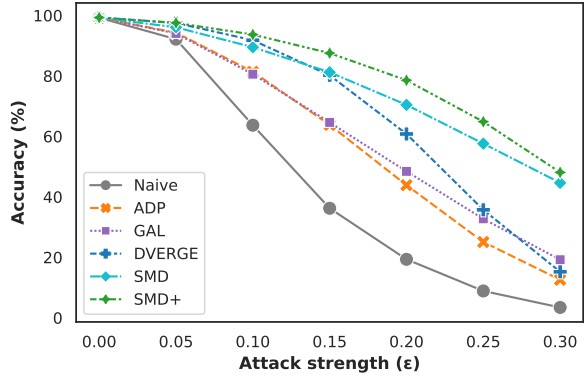

Figure 4: Accuracy vs. Attacks Strength for PGD Attacks on MNIST under adversarial training.

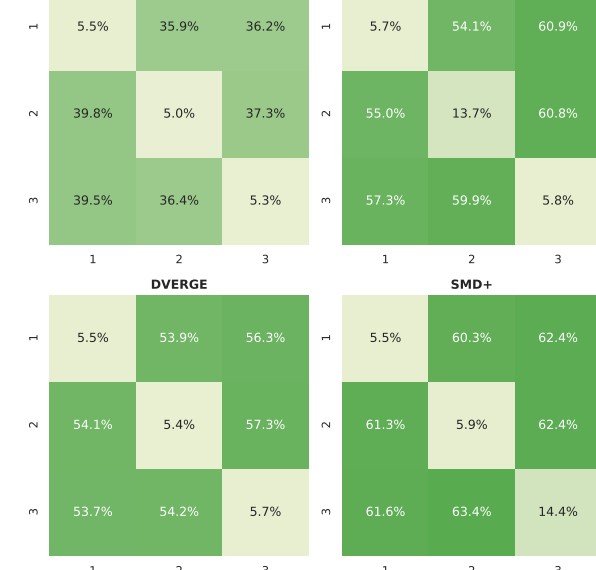

Figure 5: Transferability of PGD attacks on F-MNIST. Attacks are crafted on Y-axis members and tested on X-axis members. Higher values indicate better performance.

as described in Section 4.1. In Figure 4, we show the results for the PGD attack on MNIST dataset. In the white-box attack setting, we see major improvement for all regularizers where SMD and SMD+ consistently outperforming others. This is consistent with results from (Tramèr et al. 2018), which showed EAT to perform rather poorly in the white-box setting. In the Appendix D, we also show the results for black-box attacks.

## 5  Conclusion

In this paper, we proposed a novel diversity-promoting learning approach for the adversarial robustness of deep en-

sembles. We introduced saliency diversification measure and presented a saliency diversification learning objective. With our learning approach, we aimed at minimizing possible shared sensitivity across the ensemble members to decrease its vulnerability to adversarial attacks. Our empirical results showed a reduced transferability between ensemble members and improved performance compared to other ensemble defense methods. We also demonstrated that our approach combined with existing methods outperforms state-of-the-art ensemble algorithms in adversarial robustness.

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

# A. Additional Result-Supporting Metrics

In this section, we report the standard deviation of the results from the main paper based on 5 independent trials.

In Fig. 6 and 7, and Tab. 3 and 4, we show the results for standard deviations. As we can see from the results, SMD has higher variance than SMD+. Nonetheless, we point out that even under such variation SMD has significant gain other the comparing state-of-the-art algorithms for an attacks with high strength. In is also important to note that for the results on the MNIST and F-MNIST dataset the DVERGE method also has high variance and it is lower but comparable to the SMD. On the other hand it seems that the combination SMD+ has relatively low variance, and interestingly, in the majority of the results it is lower than both SMD and DVERGE.

We show average over 5 independent trials (as in the main paper) and the standard deviation for the transferability of the attacks between the ensemble members, which measures how likely the crafted white-box attack for one ensemble member succeeds on another. In all of the results the Y-axis represents the member from which the adversary crafts the attack (*i.e.* source), and the X-axis - the member on which the adversary transfers the attack (*i.e.* target).

The on diagonal values depict the accuracy of a particular ensemble member under a white-box attack. We see that both SMD and SMD+ models have high ensemble resilience. It appears that at some of the ensemble members the variance in the estimate for SMD is high. Interestingly, we found out that this is due to the fact that in the prediction of the SMD ensemble over 5 independent runs, we have one prediction which is quite high and thus causes this deviation. This suggest that an additional tuning of the hyperparameters for the SMD approach might lead to even better performance, which we leave it as future work.

The other (off-diagonal) values show the accuracy of the target members under transferred (black-box) attacks from the source member, here we see that the variance is on levels comparable with the baseline methods.

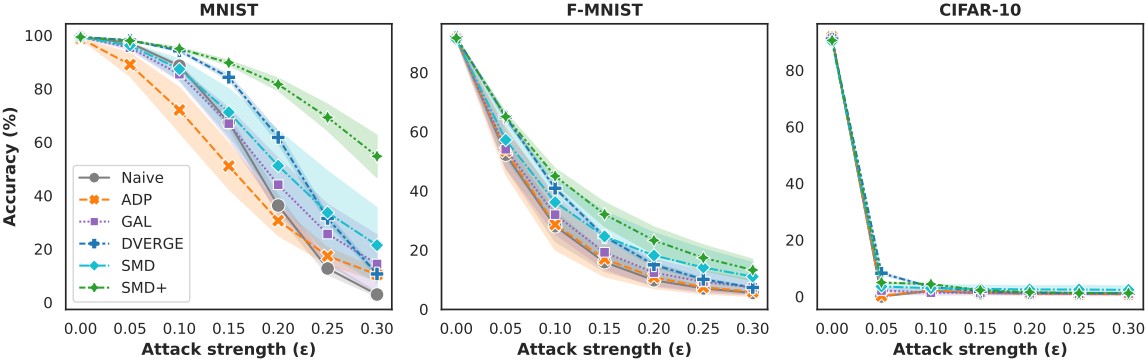

Figure 6: Accuracy vs. attacks strength for white-box PGD attacks on an ensemble of 3 LeNet-5 models for MNIST and F-MNIST and on an ensemble of 3 ReNets-20 for CIFAR-10 dataset.

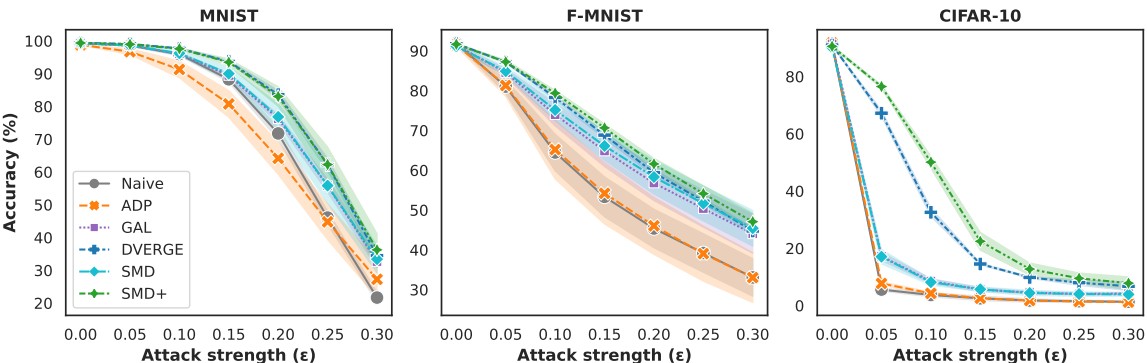

Figure 7: Accuracy vs. attacks strength for black-box PGD attacks on an ensemble of 3 LeNet-5 models for MNIST and F-MNIST and on an ensemble of 3 ReNets-20 for CIFAR-10 dataset.

| | MNIST | | | | | | F-MNIST | | | | | | CIFAR-10 | | | | | |
|---|---|---|---|---|---|---|---|---|---|---|---|---|---|---|---|---|---|---|
| | Clean | $F_{gsm}$ | R-F. | PGD | BIM | MIM | Clean | $F_{gsm}$ | R-F. | PGD | BIM | MIM | Clean | $F_{gsm}$ | R-F. | PGD | BIM | MIM |
| Naive | 0.0 | 3.5 | 1.8 | 0.7 | 0.9 | 1.4 | 0.1 | 2.2 | 1.7 | 0.4 | 0.9 | 0.7 | 0.4 | 0.6 | 0.7 | 0.3 | 0.6 | 0.5 |
| ADP | 0.1 | 8.8 | 4.3 | 2.2 | 5.6 | 4.7 | 0.3 | 2.6 | 3.5 | 1.5 | 2.1 | 1.6 | 0.1 | 0.6 | 0.8 | 0.0 | 0.0 | 0.1 |
| GAL | 0.1 | 4.4 | 1.5 | 10.9 | 9.4 | 9.3 | 0.4 | 5.5 | 2.9 | 2.5 | 3.7 | 4.3 | 0.4 | 1.2 | 1.7 | 0.6 | 0.9 | 1.9 |
| DV. | 0.0 | 3.6 | 0.9 | 1.0 | 1.6 | 2.3 | 0.1 | 1.8 | 1.6 | 0.2 | 0.5 | 0.7 | 0.1 | 0.3 | 1.4 | 0.1 | 0.1 | 0.3 |
| SMD | 0.1 | 9.3 | 1.2 | 14.0 | 17.4 | 16.6 | 0.4 | 6.4 | 3.2 | 4.7 | 6.1 | 6.1 | 0.6 | 1.1 | 1.0 | 1.3 | 0.9 | 1.4 |
| SMD+ | 0.0 | 1.3 | 1.1 | 7.9 | 3.7 | 2.2 | 0.2 | 2.6 | 2.1 | 3.6 | 4.5 | 4.2 | 0.3 | 0.4 | 2.2 | 0.2 | 0.3 | 0.2 |

Table 3: Standard deviations for white-box attacks of the magnitude $\epsilon = 0.3$ on an ensemble of 3 LeNet-5 models for MNIST and F-MNIST and on an ensemble of 3 ReNets-20 for CIFAR-10 dataset. Columns are attacks and rows are defenses employed.

| | MNIST | | | | | | F-MNIST | | | | | | CIFAR-10 | | | | | |
|---|---|---|---|---|---|---|---|---|---|---|---|---|---|---|---|---|---|---|
| | Clean | $F_{gsm}$ | R-F. | PGD | BIM | MIM | Clean | $F_{gsm}$ | R-F. | PGD | BIM | MIM | Clean | $F_{gsm}$ | R-F. | PGD | BIM | MIM |
| Naive | 0.0 | 1.9 | 0.8 | 1.5 | 1.3 | 0.9 | 0.1 | 2.4 | 2.6 | 4.7 | 3.4 | 1.8 | 0.4 | 0.5 | 1.3 | 0.2 | 0.1 | 0.1 |
| ADP | 0.1 | 6.0 | 5.8 | 5.4 | 5.4 | 4.7 | 0.3 | 3.5 | 4.4 | 6.2 | 4.5 | 2.7 | 0.1 | 0.8 | 0.6 | 0.0 | 0.0 | 0.2 |
| GAL | 0.1 | 1.0 | 1.7 | 1.9 | 2.3 | 2.1 | 0.4 | 4.0 | 3.9 | 4.9 | 3.8 | 3.1 | 0.4 | 0.4 | 0.4 | 0.4 | 0.1 | 1.2 |
| DV. | 0.0 | 0.7 | 0.5 | 1.6 | 1.2 | 0.5 | 0.1 | 0.9 | 1.1 | 0.8 | 0.5 | 0.7 | 0.1 | 0.4 | 1.1 | 1.5 | 0.3 | 0.3 |
| SMD | 0.1 | 3.1 | 2.4 | 4.1 | 4.0 | 2.6 | 0.4 | 4.2 | 4.0 | 4.5 | 3.8 | 3.1 | 0.6 | 0.3 | 0.5 | 0.6 | 0.1 | 0.2 |
| SMD+ | 0.0 | 3.6 | 1.5 | 4.9 | 4.2 | 2.6 | 0.2 | 2.2 | 1.8 | 2.1 | 1.2 | 1.5 | 0.3 | 0.2 | 1.7 | 2.2 | 2.0 | 0.3 |

Table 4: Standard deviations for black-box attacks of the magnitude $\epsilon = 0.3$ on an ensemble of 3 LeNet-5 models for MNIST and F-MNIST and on an ensemble of 3 ReNets-20 for CIFAR-10 dataset. Columns are attacks and rows are defenses employed.

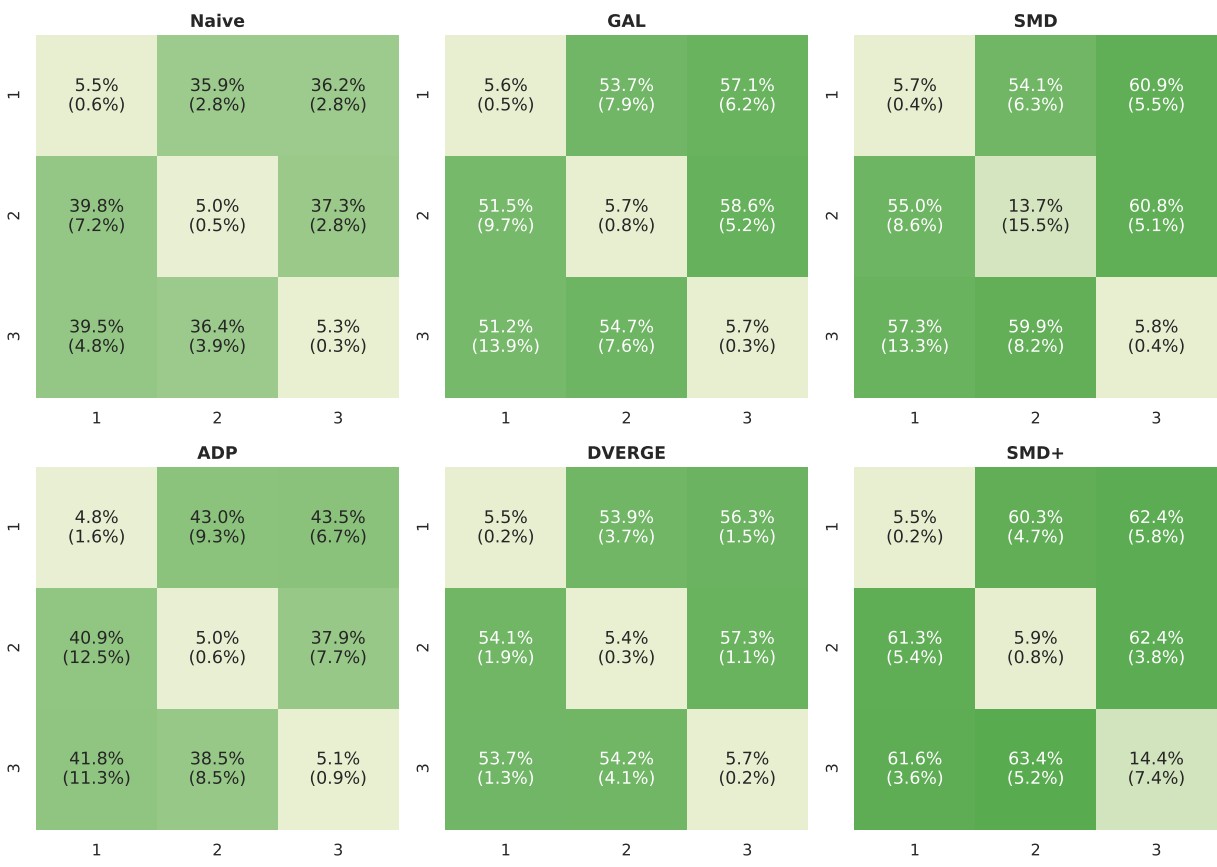

Figure 8: Transferability of PGD attacks on F-MNIST. Attacks are crafted on Y-axis members and tested on X-axis members. Higher values indicate better performance. Standard deviations are in parenthesis.

# B. Results for Additional Attacks

In this section, we show results for additional attacks in with-box and black-box setting. Namely, in addition to PGD attacks shown in the main text we present FGSM, R-FGMS, MIM and BIM attacks here.

In Fig. 9, 10, 11, 12, 13, 14, 15, 16, we show the results. Similarly as in the main paper, we can see gains in performance for our SMD approach compared to the existing methods. The results appear to be consistent with those presented in the main text with SMD and SMD+ methods outperforming the baselines in most cases.

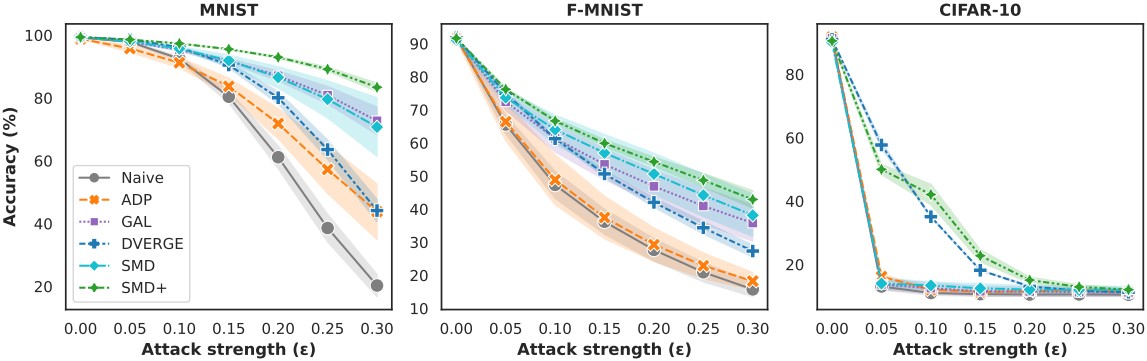

Figure 9: Accuracy vs. attacks strength for white-box FGSM attacks on an ensemble of 3 LeNet-5 models for MNIST and F-MNIST and on an ensemble of 3 ReNets-20 for CIFAR-10 dataset.

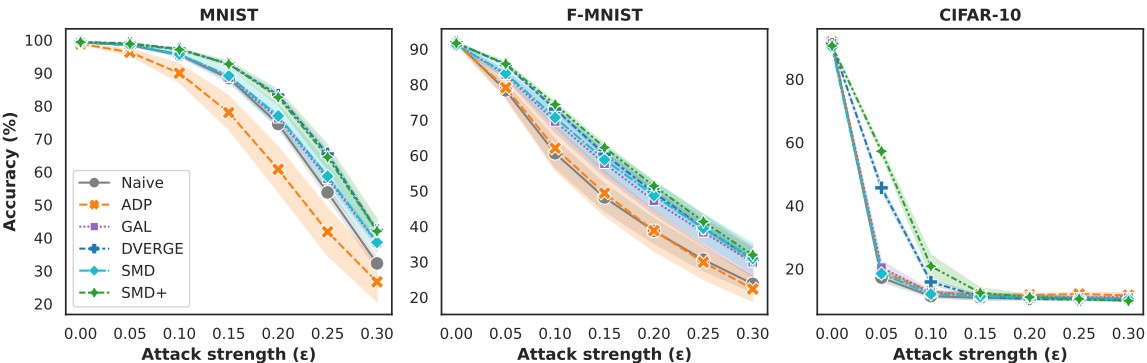

Figure 10: Accuracy vs. attacks strength for black-box FGSM attacks on an ensemble of 3 LeNet-5 models for MNIST and F-MNIST and on an ensemble of 3 ReNets-20 for CIFAR-10 dataset.

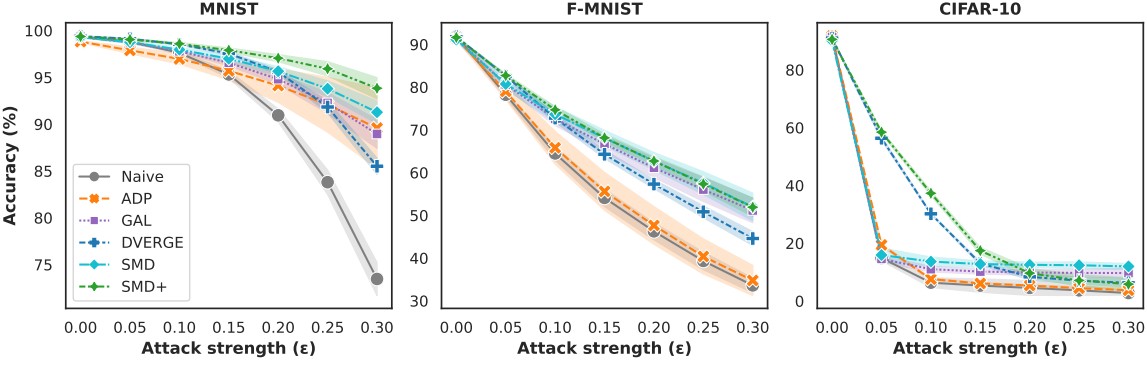

Figure 11: Accuracy vs. attacks strength for white-box R-FGSM attacks on an ensemble of 3 LeNet-5 models for MNIST and F-MNIST and on an ensemble of 3 ReNets-20 for CIFAR-10 dataset.

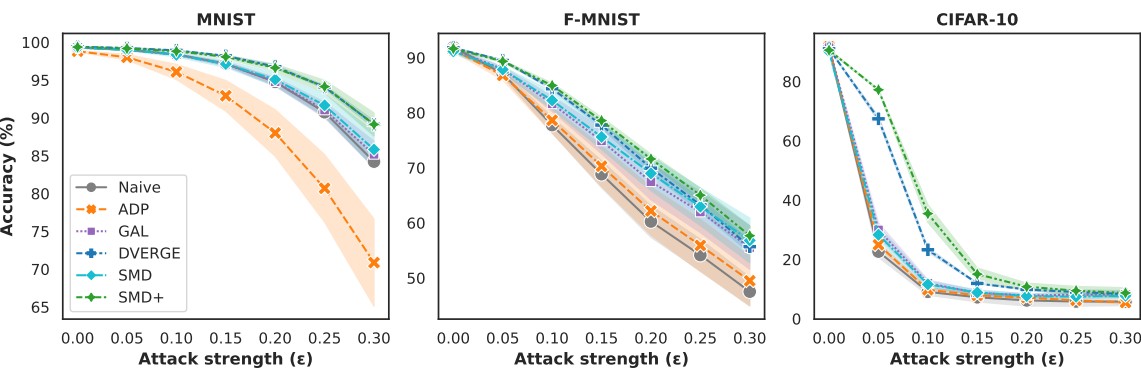

Figure 12: Accuracy vs. attacks strength for black-box R-FGSM attacks on an ensemble of 3 LeNet-5 models for MNIST and F-MNIST and on an ensemble of 3 ReNets-20 for CIFAR-10 dataset.

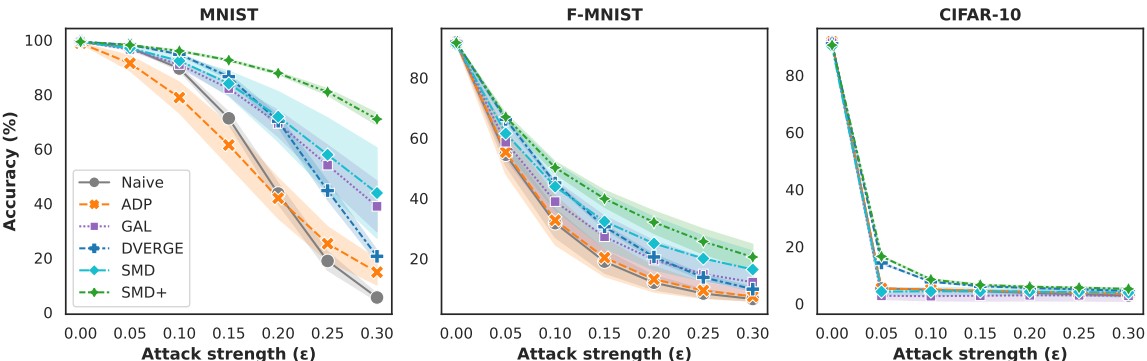

Figure 13: Accuracy vs. attacks strength for white-box MIM attacks on an ensemble of 3 LeNet-5 models for MNIST and F-MNIST and on an ensemble of 3 ReNets-20 for CIFAR-10 dataset.

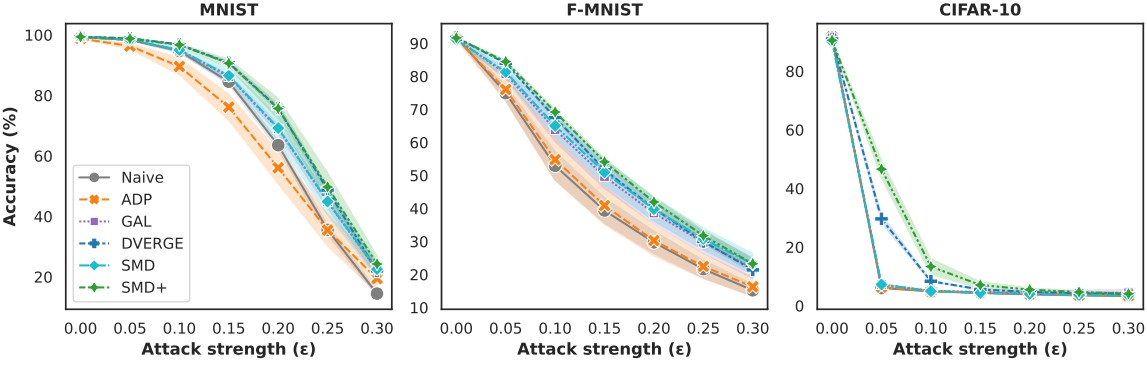

Figure 14: Accuracy vs. attacks strength for black-box MIM attacks on an ensemble of 3 LeNet-5 models for MNIST and F-MNIST and on an ensemble of 3 ReNets-20 for CIFAR-10 dataset.

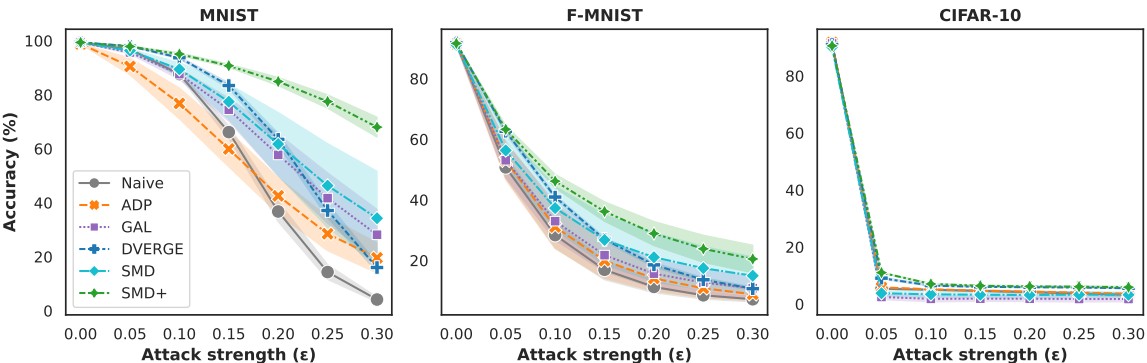

Figure 15: Accuracy vs. attacks strength for white-box BIM attacks on an ensemble of 3 LeNet-5 models for MNIST and F-MNIST and on an ensemble of 3 ReNets-20 for CIFAR-10 dataset.

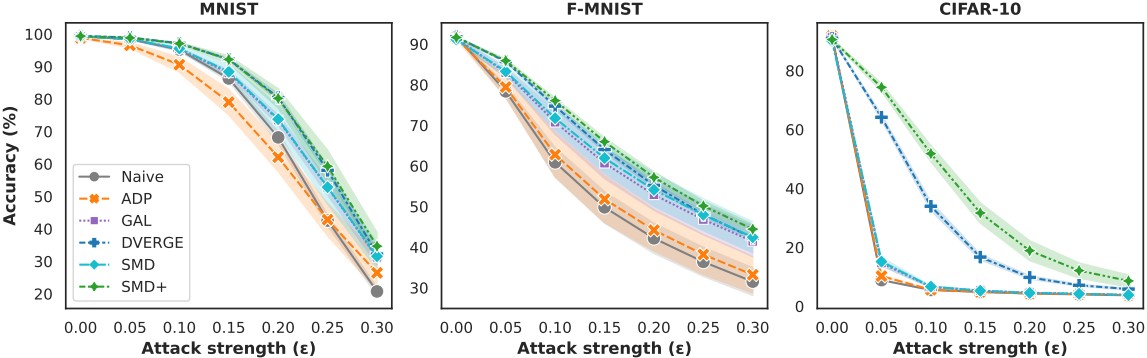

Figure 16: Accuracy vs. attacks strength for black-box BIM attacks on an ensemble of 3 LeNet-5 models for MNIST and F-MNIST and on an ensemble of 3 ReNets-20 for CIFAR-10 dataset.

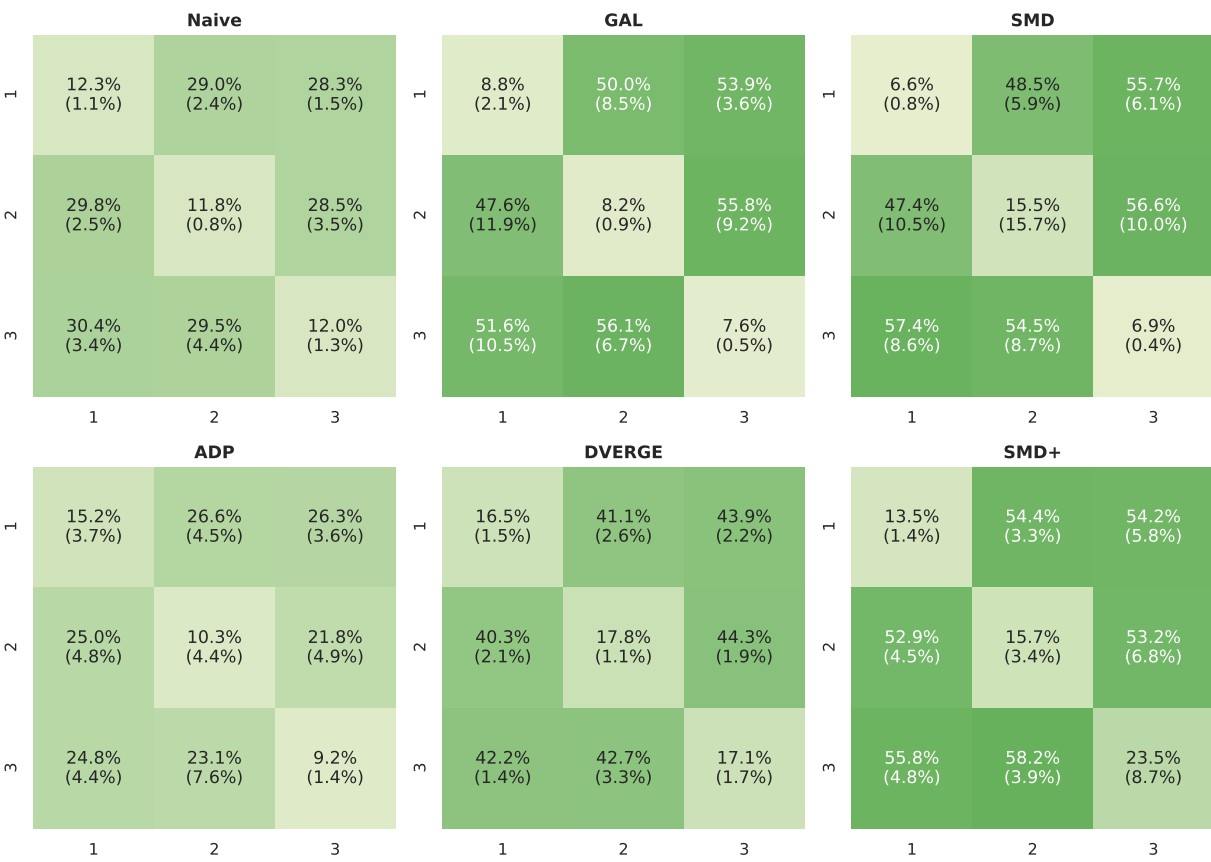

Figure 17: Transferability of FGSM attacks on F-MNIST. Attacks are crafted on Y-axis members and tested on X-axis members. Higher values indicate better performance. Standard deviations are in parenthesis.

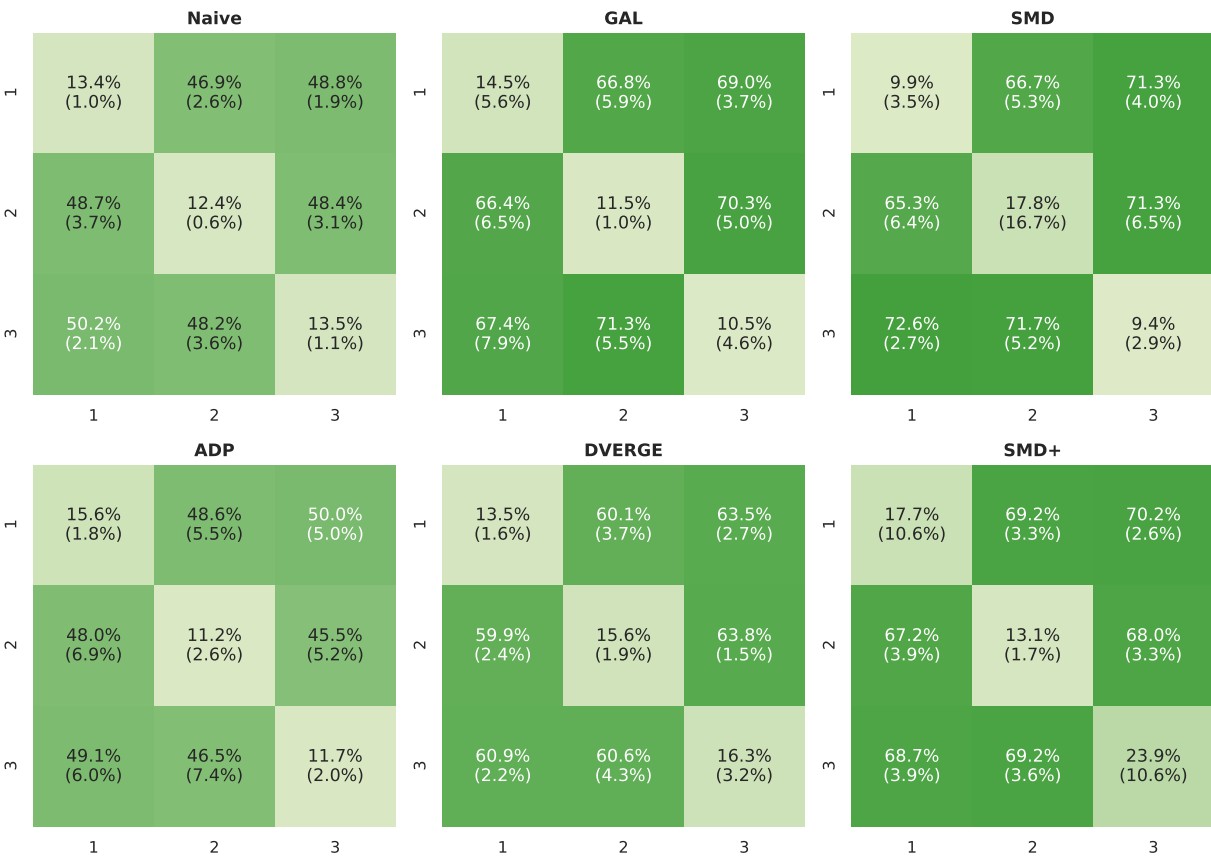

Figure 18: Transferability of R-FGSM attacks on F-MNIST. Attacks are crafted on Y-axis members and tested on X-axis members. Higher values indicate better performance. Standard deviations are in parenthesis.

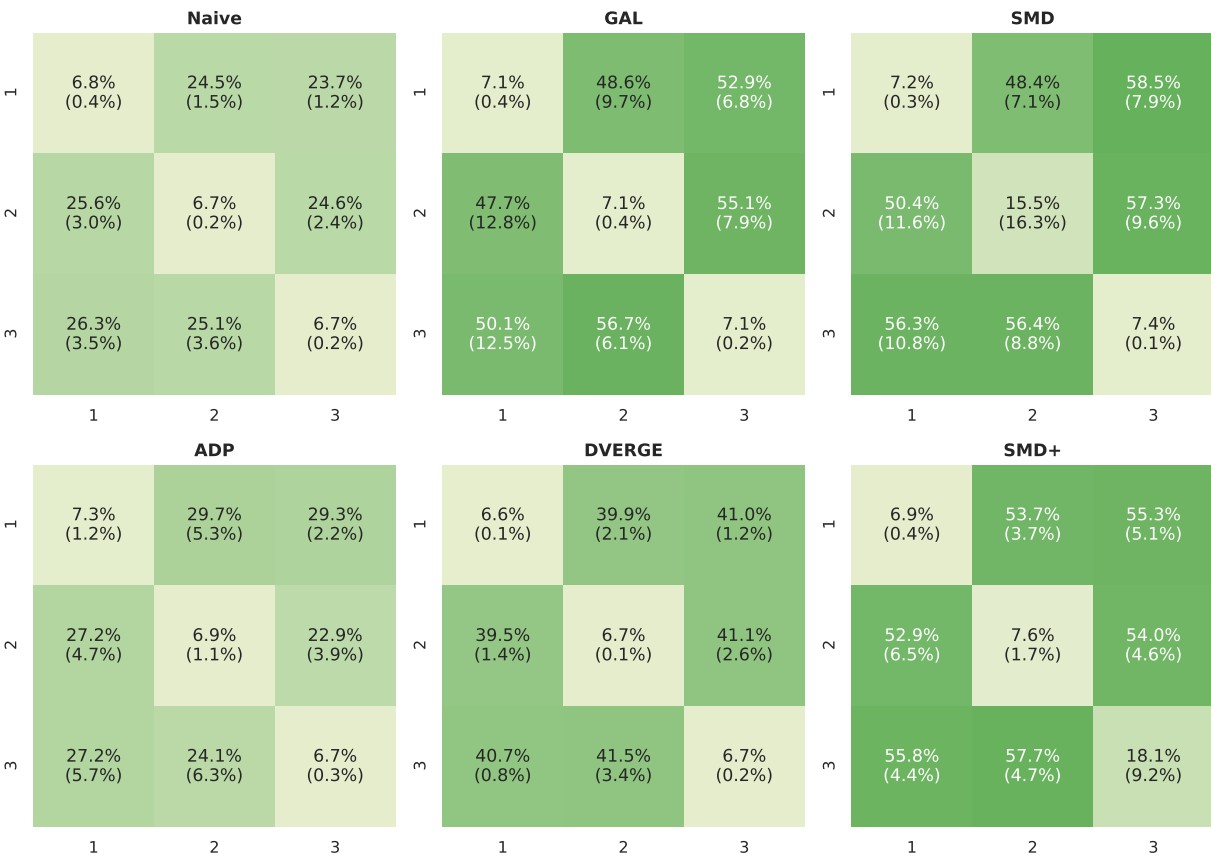

Figure 19: Transferability of MIM attacks on F-MNIST. Attacks are crafted on Y-axis members and tested on X-axis members. Higher values indicate better performance. Standard deviations are in parenthesis.

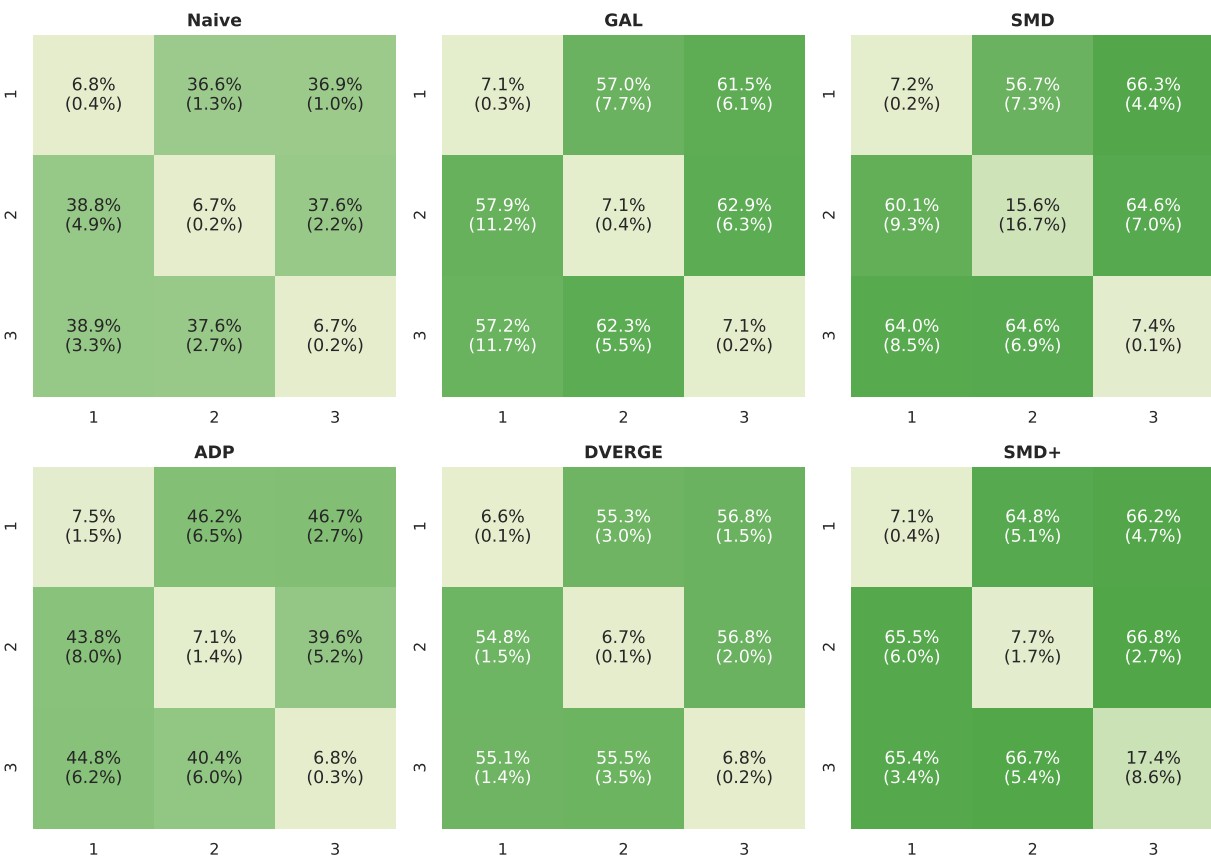

Figure 20: Transferability of BIM attacks on F-MNIST. Attacks are crafted on Y-axis members and tested on X-axis members. Higher values indicate better performance. Standard deviations are in parenthesis.

# C. Impact of the Number of Ensemble Members

In this section, we show the results for ensembles of 5 and 8 members using the MNIST, F-MNIST and CIFAR-10 datasets under withe-box and black-box attacks. For MNIST and F-MNIST we use 5 seeds for the evaluation, while we use 3 seed for CIFAR-10 due to ResNet-20 being much slower to train.

In Fig. 21 and 22, and Tab. 5 and 6, we can see that when we use an ensemble of 5 members, we sill have high accuracy in the black-box and white-box attack setting. Moreover in the black-box setting, we have better results for most of the attacks, while in the black-box settings we have still have better results for almost all of the attacks compared to the state-of-the-art methods.

The results for 8-member ensembles are shown in In Fig. 23 and 24, and Tab. 7 and 8. These results are also consistent in terms of the performance gains for the SMD and SMD+ methods compared with the results for the 3 and 5-member ensembles.

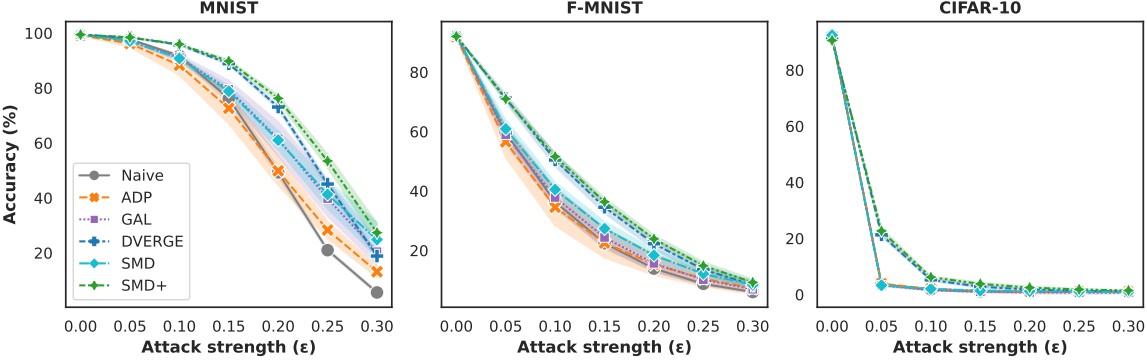

Figure 21: Accuracy vs. attacks strength for white-box PGD attacks on an ensemble of 5 LeNet-5 models for MNIST and F-MNIST and on an ensemble of 5 ReNets-20 for CIFAR-10 dataset.

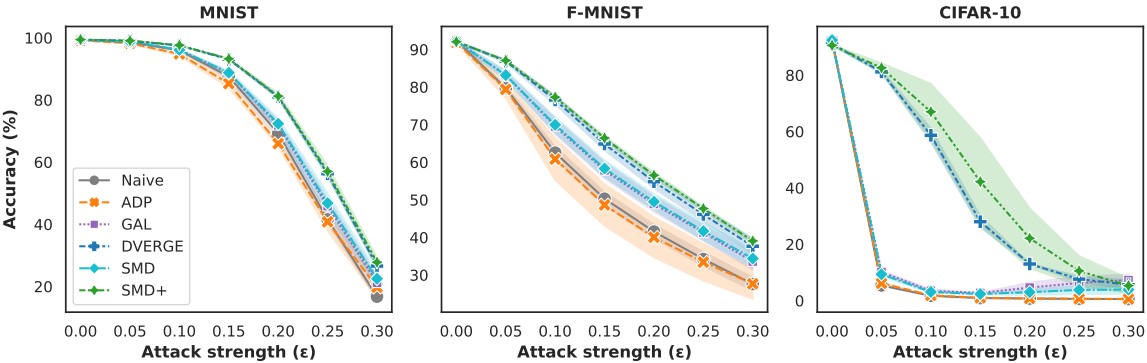

Figure 22: Accuracy vs. attacks strength for black-box PGD attacks on an ensemble of 5 LeNet-5 models for MNIST and F-MNIST and on an ensemble of 5 ReNets-20 for CIFAR-10 dataset.

| | MNIST | | | | | | F-MNIST | | | | | | CIFAR-10 | | | | | |
| --- | --- | --- | --- | --- | --- | --- | --- | --- | --- | --- | --- | --- | --- | --- | --- | --- | --- | --- |
| | Clean | $F_{gsm}$ | R-F. | PGD | BIM | MIM | Clean | $F_{gsm}$ | R-F. | PGD | BIM | MIM | Clean | $F_{gsm}$ | R-F. | PGD | BIM | MIM |
| Naive | 99.4 | 24.7 | 79.1 | 5.6 | 7.8 | 8.5 | **92.4** | 18.0 | 37.5 | 6.0 | 8.5 | 7.6 | 92.3 | 10.7 | 2.5 | 1.0 | 3.1 | 2.7 |
| ADP | 99.2 | 46.2 | 89.0 | 13.2 | 24.0 | 18.7 | 91.9 | 19.3 | 37.4 | 7.2 | 11.4 | 9.1 | 92.2 | 11.5 | 4.1 | 0.9 | 3.2 | 2.8 |
| GAL | 99.4 | **81.7** | 91.0 | 20.4 | **47.1** | **54.6** | 92.3 | **37.8** | 50.8 | 6.9 | 12.8 | 12.7 | 92.4 | 10.1 | **9.1** | 0.7 | 1.0 | 1.6 |
| DV. | 99.4 | 48.2 | 88.5 | 18.9 | 27.8 | 28.2 | 92.1 | 26.8 | 47.1 | 8.3 | 13.6 | 12.3 | 91.1 | **12.3** | 5.1 | 1.1 | 5.6 | 5.0 |
| SMD | 99.4 | 75.2 | 91.8 | 24.8 | 41.9 | 49.3 | 92.2 | 37.5 | **51.2** | 8.4 | 15.4 | **15.1** | **92.4** | 10.7 | 6.9 | 0.9 | 1.3 | 0.8 |
| SMD+ | **99.4** | 67.6 | **92.3** | **27.4** | 43.6 | 46.0 | 92.0 | 32.4 | 50.7 | **9.2** | **16.4** | 14.4 | 90.6 | 11.2 | 4.4 | **1.5** | **6.1** | **5.7** |

Table 5: White-box attacks of the magnitude $\epsilon = 0.3$ on an ensemble of 5 LeNet-5 models for MNIST and F-MNIST and on an ensemble of 5 ReNets-20 for CIFAR-10 dataset. Columns are attacks and rows are defenses employed.

| | MNIST | | | | | | F-MNIST | | | | | | CIFAR-10 | | | | | |
|---|---|---|---|---|---|---|---|---|---|---|---|---|---|---|---|---|---|---|
| | Clean | $F_{gsm}$ | R-F. | PGD | BIM | MIM | Clean | $F_{gsm}$ | R-F. | PGD | BIM | MIM | Clean | $F_{gsm}$ | R-F. | PGD | BIM | MIM |
| Naive | 99.4 | 31.1 | 84.0 | 16.7 | 17.2 | 12.6 | **92.4** | 23.5 | 46.7 | 27.6 | 27.1 | 13.0 | 92.3 | 10.9 | 5.6 | 0.5 | 2.7 | 2.2 |
| ADP | 99.2 | 27.3 | 78.3 | 19.7 | 19.6 | 14.4 | 91.9 | 22.9 | 46.2 | 27.7 | 28.1 | 14.1 | 92.2 | 11.3 | 5.7 | 0.6 | 2.7 | 2.3 |
| GAL | 99.4 | 35.9 | 84.6 | 21.2 | 21.5 | 16.7 | 92.3 | 26.7 | 50.6 | 33.6 | 32.8 | 15.6 | 92.4 | 10.7 | **9.5** | **7.3** | 2.7 | **3.1** |
| DV. | 99.4 | 39.1 | 88.2 | 26.6 | 26.2 | 18.3 | 92.1 | 28.4 | 54.2 | 37.6 | 36.8 | 17.3 | 91.1 | 10.3 | 7.1 | 5.6 | 6.2 | 2.4 |
| SMD | 99.4 | 35.5 | 84.9 | 22.5 | 23.2 | 17.9 | 92.2 | 28.0 | 51.3 | 34.4 | 34.3 | 17.3 | **92.4** | **11.4** | 8.6 | 3.9 | 2.7 | 2.1 |
| SMD+ | **99.4** | **41.2** | **88.4** | **27.8** | **27.5** | **20.0** | 92.0 | **29.7** | **55.1** | **39.0** | **38.4** | **18.7** | 90.6 | 10.1 | 5.4 | 5.3 | **10.7** | 2.3 |

Table 6: Black-box attacks of the magnitude $\epsilon = 0.3$ on an ensemble of 5 LeNet-5 models for MNIST and F-MNIST and on an ensemble of 5 ReNets-20 for CIFAR-10 dataset. Columns are attacks and rows are defenses employed.

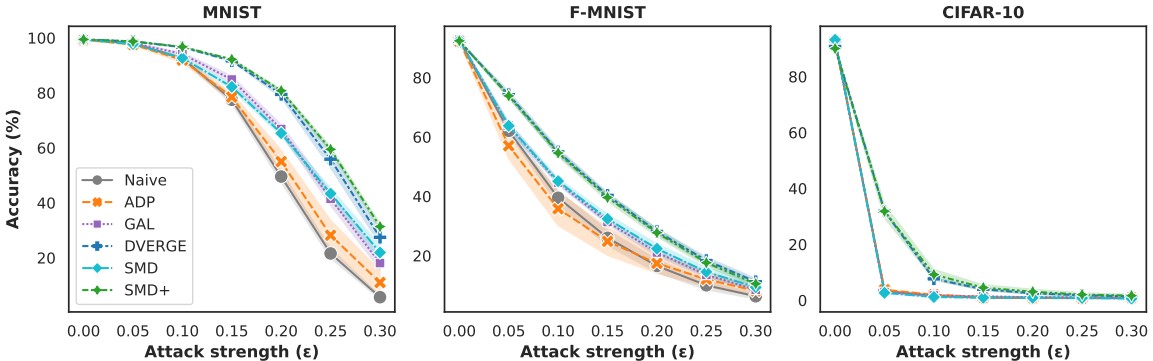

Figure 23: Accuracy vs. attacks strength for white-box PGD attacks on an ensemble of 8 LeNet-5 models for MNIST and F-MNIST and on an ensemble of 8 ReNets-20 for CIFAR-10 dataset.

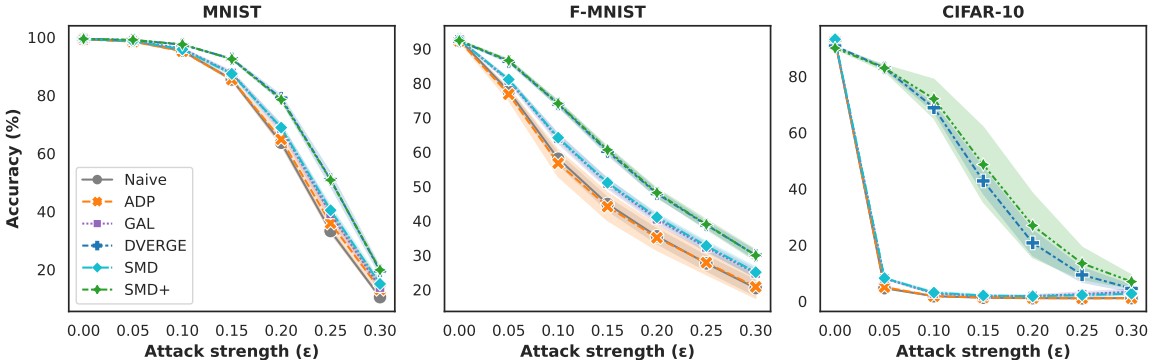

Figure 24: Accuracy vs. attacks strength for black-box PGD attacks on an ensemble of 8 LeNet-5 models for MNIST and F-MNIST and on an ensemble of 8 ReNets-20 for CIFAR-10 dataset.

| | MNIST | | | | | | F-MNIST | | | | | | CIFAR-10 | | | | | |
|---|---|---|---|---|---|---|---|---|---|---|---|---|---|---|---|---|---|---|
| | Clean | $F_{gsm}$ | R-F. | PGD | BIM | MIM | Clean | $F_{gsm}$ | R-F. | PGD | BIM | MIM | Clean | $F_{gsm}$ | R-F. | PGD | BIM | MIM |
| Naive | 99.4 | 22.8 | 78.9 | 5.7 | 8.1 | 8.1 | 92.7 | 16.8 | 39.0 | 6.3 | 8.8 | 7.2 | 92.8 | 10.8 | 1.5 | 0.8 | 2.8 | 2.5 |
| ADP | 99.3 | 38.3 | 83.8 | 11.0 | 18.1 | 15.4 | 92.3 | 15.9 | 37.4 | 8.2 | 11.7 | 7.3 | 92.7 | 11.3 | 2.4 | 0.8 | 3.2 | 2.8 |
| GAL | 99.4 | 59.4 | 90.1 | 18.1 | 28.9 | 31.3 | **92.7** | 32.0 | 50.5 | 8.5 | 14.6 | 12.0 | 92.9 | 10.0 | 7.8 | 0.7 | 1.6 | 0.5 |
| DV. | 99.4 | 54.7 | 90.5 | 27.5 | 37.8 | 34.7 | 92.3 | 28.6 | 47.4 | **11.2** | **18.4** | 14.9 | 90.8 | **11.9** | 3.2 | 1.4 | 5.7 | 5.4 |
| SMD | 99.4 | **73.1** | 91.5 | 21.9 | 40.4 | **43.8** | 92.6 | **37.4** | **52.3** | 9.4 | 18.2 | **15.7** | **93.2** | 9.8 | **8.4** | 0.6 | 1.2 | 0.5 |
| SMD+ | **99.5** | 60.3 | **91.8** | **31.4** | **43.2** | 40.2 | 92.4 | 29.5 | 48.5 | 10.6 | 17.9 | 14.6 | 90.1 | 11.9 | 4.9 | **1.7** | **6.2** | **5.9** |

Table 7: White-box attacks of the magnitude $\epsilon = 0.3$ on an ensemble of 8 LeNet-5 models for MNIST and F-MNIST and on an ensemble of 8 ReNets-20 for CIFAR-10 dataset. Columns are attacks and rows are defenses employed.

| | MNIST | | | | | | F-MNIST | | | | | | CIFAR-10 | | | | | |
|---|---|---|---|---|---|---|---|---|---|---|---|---|---|---|---|---|---|---|
| | Clean | $F_{gsm}$ | R-F. | PGD | BIM | MIM | Clean | $F_{gsm}$ | R-F. | PGD | BIM | MIM | Clean | $F_{gsm}$ | R-F. | PGD | BIM | MIM |
| Naive | 99.4 | 26.4 | 82.0 | 10.5 | 11.5 | 9.5 | 92.7 | 22.5 | 43.7 | 20.4 | 21.2 | 10.8 | 92.8 | 10.9 | 2.5 | 1.1 | 3.1 | 2.5 |
| ADP | 99.3 | 27.9 | 81.2 | 13.2 | 13.8 | 11.7 | 92.3 | 21.3 | 43.5 | 20.8 | 22.4 | 11.4 | 92.7 | **11.4** | 2.7 | 1.1 | 3.2 | 2.6 |
| GAL | 99.4 | 33.2 | 83.9 | 13.8 | 14.8 | 13.1 | **92.7** | 25.8 | 47.5 | 24.7 | 25.2 | 13.2 | 92.9 | 10.2 | **8.1** | 3.4 | 3.1 | 2.6 |
| DV. | 99.4 | 36.9 | **87.9** | 19.6 | 20.0 | 16.2 | 92.3 | **28.6** | 51.0 | **30.0** | **30.7** | **15.3** | 90.8 | 11.0 | 4.7 | 4.6 | 9.0 | 2.6 |
| SMD | 99.4 | 33.8 | 83.8 | 15.0 | 16.0 | 14.1 | 92.6 | 26.1 | 47.9 | 25.1 | 25.8 | 13.5 | **93.2** | 10.1 | 7.8 | 2.7 | 3.0 | 2.5 |
| SMD+ | **99.5** | **37.8** | 87.3 | **19.9** | **20.2** | **16.6** | 92.4 | 28.6 | **51.0** | 30.0 | 30.5 | 15.0 | 90.1 | 10.5 | 6.8 | **7.0** | **12.4** | **2.7** |

Table 8: Black-box attacks of the magnitude $\epsilon = 0.3$ on an ensemble of 8 LeNet-5 models for MNIST and F-MNIST and on an ensemble of 8 ReNets-20 for CIFAR-10 dataset. Columns are attacks and rows are defenses employed.

# D. Additional Adversarial Training Results

In this section, we also present an additional results where we complement the results in our paper with the results about the variance. In addition, we also show results for adversarial training and black-box attacks. We also show results for the F-MNIST data set in black-box and white-box setting.

In the white-box attack setting for the two datasets, we see major improvement for all regularizers where SMD and SMD+ consistently outperforming others. Considering the results for in the black-box setting we do not have gains. Again this is consistent with results from (Tramèr et al. 2018).

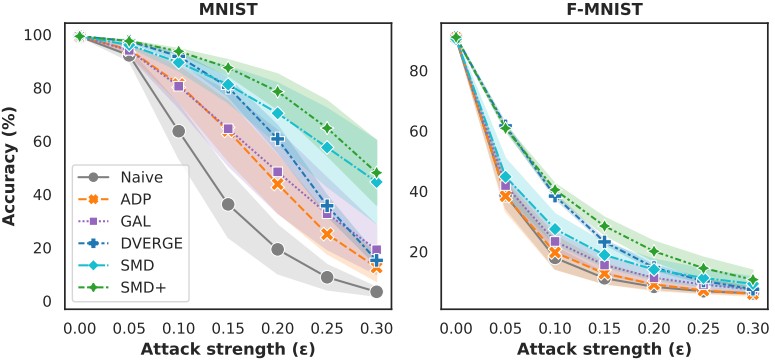

Figure 25: Accuracy vs. attacks strength for white-box PGD attacks on an ensemble of 3 LeNet-5 models with adversarial training for MNIST and F-MNIST datasets.

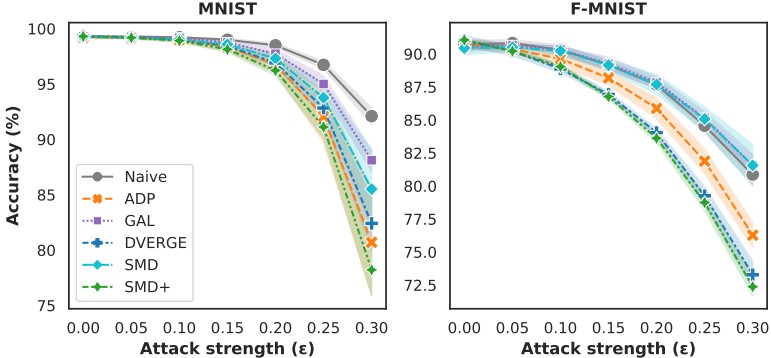

Figure 26: Accuracy vs. attacks strength for black-box PGD attacks on an ensemble of 3 LeNet-5 models with adversarial training for MNIST and F-MNIST datasets.

| | | MNIST | | | | | F-MNIST | | | | |
| | Clean | $F_{gsm}$ | R-F. | PGD | BIM | MIM | Clean | $F_{gsm}$ | R-F. | PGD | BIM | MIM |
|---|---|---|---|---|---|---|---|---|---|---|---|---|
| Naive | 99.2 | 32.9 | 76.5 | 3.4 | 4.9 | 6.0 | 90.7 | 13.2 | 26.2 | 6.2 | 7.6 | 7.2 |
| ADP | 99.2 | 50.8 | 84.3 | 12.6 | 20.7 | 19.7 | 90.8 | 16.2 | 29.3 | 5.9 | 8.4 | 7.4 |
| GAL | 99.3 | 80.1 | 91.9 | 19.2 | 38.2 | 44.8 | 90.5 | **39.5** | 41.0 | 7.4 | 10.9 | 13.0 |
| DV. | **99.3** | 65.2 | 90.0 | 15.2 | 26.2 | 31.7 | 91.0 | 26.6 | 44.2 | 7.5 | 11.2 | 10.5 |
| SMD | 99.3 | 81.7 | 91.4 | 44.6 | 60.5 | 63.6 | 90.4 | 38.7 | 44.7 | 9.3 | 13.4 | 15.3 |
| SMD+ | 99.3 | **85.1** | **94.3** | **48.1** | **64.3** | **66.3** | **91.1** | 39.1 | **46.4** | **10.7** | **17.8** | **17.4** |

Table 9: White-box attacks of the magnitude $\epsilon = 0.3$ on an ensemble of 3 LeNet-5 models with adversarial training for MNIST and F-MNIST datasets. Columns are attacks and rows are defenses employed.

|  | MNIST | | | | | | F-MNIST | | | | | |
|---|---|---|---|---|---|---|---|---|---|---|---|---|
|  | Clean | $F_{gsm}$ | R-F. | PGD | BIM | MIM | Clean | $F_{gsm}$ | R-F. | PGD | BIM | MIM |
| Naive | 99.2 | **85.4** | **97.6** | **92.1** | **90.9** | **84.4** | 90.7 | 62.3 | 77.7 | 80.9 | 84.0 | 69.5 |
| ADP | 99.2 | 71.3 | 95.3 | 80.7 | 79.4 | 66.7 | 90.8 | 57.0 | 75.9 | 76.3 | 82.1 | 63.7 |
| GAL | 99.3 | 81.4 | 96.9 | 88.1 | 87.4 | 78.2 | 90.5 | 63.1 | 78.4 | **81.6** | **85.0** | 70.8 |
| DV. | **99.3** | 76.9 | 96.2 | 82.4 | 79.4 | 68.2 | 91.0 | 52.8 | 74.2 | 73.3 | 74.8 | 52.2 |
| SMD | 99.3 | 78.9 | 96.7 | 85.5 | 84.3 | 74.4 | 90.4 | **63.9** | **78.6** | 81.6 | 84.9 | **71.1** |
| SMD+ | 99.3 | 73.4 | 96.1 | 78.2 | 76.1 | 63.1 | **91.1** | 51.0 | 72.6 | 72.4 | 75.2 | 52.7 |

Table 10: Black-box attacks of the magnitude $\epsilon = 0.3$ on an ensemble of 3 LeNet-5 models with adversarial training for MNIST and F-MNIST datasets. Columns are attacks and rows are defenses employed.