# OpenReview forum: "Saliency Diversified Deep Ensemble for Robustness to Adversaries"
_AAAI.org/2022/Workshop/AdvML — AAAI-22 AdvML Workshop LongPaper_

### Official Review · Reviewer_UNi9 · 2021-11-29
**Saliency Diversified Deep Ensemble for Robustness to Adversaries**

**Rating:** 7
**Confidence:** 4

**Review:**

This paper proposes a diversity-promoting learning approach for the deep ensembles, which promotes saliency map diversity (SMD) on ensemble members to prevent the attacker from targeting all ensemble members by introducing an additional term. Thus it can improve ensemble robustness to adversaries. However, some concerns are also listed as follows:

Adversarial training is a currently popular and effective method. What is the effect of this method on CIFAR-10 under adversarial training? And what are the corresponding time-consuming results? Besides, AutoAttack[1] can be involved in this paper for white-box evaluation.

[1] Reliable evaluation of adversarial robustness with an ensemble of diverse parameter-free attacks.

---

### Official Review · Reviewer_vqvD · 2021-11-30
**Review of Paper4**

**Rating:** 6
**Confidence:** 4

**Review:**

**Summary of the paper:**

The authors propose a novel diversity-promoting learning approach for the deep ensembles to overcome the shared vulnerabilities in its members. Experiments on  MNIST, Fashion-MNIST, and CIFAR-10 are conducted to demonstrate the efficacy of the proposed methods.

**Main Review:**

Strength:
* The proposed method is simple and achieves the goal of improving the adversarial robustness of deep ensembles.
* Several ablation studies are presented to illustrate the proposed method.
Weakness:
* The saliency diversification learning objective may damage the interpretability of the DNNs.
* Why not take the large-scale data set (ImageNet) into consideration to evaluate the effectiveness of the proposed method.
* In Table. 1, the experiments on CIFAR-10 may show that the efficacy of the proposed method is limited.
* You can make an ablation study on CIFAR-10 compared with adversarial training (with small perturbation like $\epsilon=2/255$). Maybe adversarial training performs better on vanilla accuracy and robust accuracy.

---

### Decision · Program_Chairs · 2021-12-01

**Decision:**

Accept (Long Paper)

**Comment:**

Both reviewers agree to accept this paper.